# Smart and Age-Friendly Cities in Russia: An Exploratory Study of Attitudes, Perceptions, Quality of Life and Health Information Needs

**DOI:** 10.3390/ijerph17249212

**Published:** 2020-12-09

**Authors:** Liliya Eugenevna Ziganshina, Ekaterina V. Yudina, Liliya I. Talipova, Guzel N. Sharafutdinova, Rustem N. Khairullin

**Affiliations:** 1Interregional Clinical Diagnostic Centre (ICDC), The Ministry of Health of the Republic of Tatarstan, 12A Karbysheva Street, 420101 Kazan, Russia; tli13@mail.ru (L.I.T.); guzel_792@mail.ru (G.N.S.); icdc@icdc.ru (R.N.K.); 2Federal State Budgetary Educational Institution of Continuing Professional Education “Russian Medical Academy of Continuing Professional Education”, The Ministry of Health of the Russian Federation (RMANPO), 2/1, Barrikadnaya Street, 123995 Moscow, Russia; 3Department of Pharmacology, Kazan State Medical University (KSMU), The Ministry of Health of the Russian Federation, 49 Butlerov Street, 420012 Kazan, Russia; 4Department of Medicinal Chemistry, Kazan Federal University (KFU), The Ministry of Science and Higher Education of the Russian Federation, 18 Kremlevskaya Street, 420008 Kazan, Russia; 5Children’s Hospital N 1 of the City of Kazan, 125a Dekabristov Street, 420034 Kazan, Russia; ekaterina.v.yudina@mail.ru

**Keywords:** Kazan, Russia, health information, quality of life, ageing, ageism, Cochrane, evidence-based, medicines, consumers, awareness, age-friendly cities

## Abstract

In Russia, initiatives for healthy ageing have been growing over the last two decades; however, none use an evidence-based (EB) approach. It is proposed that Kazan, a city with a population of over a million in the European part of Russia, has good chances of moving towards age-friendliness and contributing to raising awareness about healthy ageing through Cochrane evidence. One of the eight essential features of age-friendly cities by the World Health Organisation (WHO) directly points to health services. This exploratory study assesses the health information needs of the ageing population of Kazan and the challenges people face in improving their health and longevity. Survey data were used from 134 participants, patients, caregivers and healthcare providers of the Interregional Clinical Diagnostic Centre (ICDC), aged from 30 to over 80 years, and potential associations of the studied parameters with age, gender, quality of life and other characteristics were analysed. Older people (60+) were less positive about their quality of life, took medicines more often on a daily basis (10/16 compared to 29/117 of people under 60), encountered problems with ageing (9/16 compared to 21/117 of people under 60) and rated their quality of life as unsatisfactory (4/14 compared to 9/107 of people under 60). Awareness of EB approaches and Cochrane was higher within health professions (evidence-based medicine: 42/86 vs. 13/48; Cochrane: 32/86 vs. 2/48), and health information needs did not differ between age or gender groups or people with a satisfactory and unsatisfactory quality of life. The minority (10%—13/134) were aware of ageism without age or gender differences. The low awareness calls for the need of Cochrane intervention both for consumers and those in the health profession to raise awareness to contribute to Kazan moving towards an age-friendly city.

## 1. Introduction

The strive for healthy ageing is universal. Healthy ageing is defined by the WHO as “*maintaining the functional ability that allows you to do the things you value*”, “*preserving physical and mental capacity*” in an accessible and supportive environment for older people. It is not surprising that one of the eight essential domains of age-friendly cities by the WHO deals with health services. Raising evidence-based awareness campaigns about ageing is the most important action, mandated by the WHO, in order to ensure that societies become more just and fairer with institutions becoming stronger to benefit the ageing population [1]. Further development of the WHO’s age-friendly framework was proposed recently with the focus on the role of technology in a modern urban environment [2].

In Russia, the proportion of the population over the working age reached one-quarter of its population in 2018, according to the Federal State Statistics Service [3]. At that time, the working age was declared as over 60 years for men and over 55 years for women until 2019 when the retirement age was raised. This demographic shift caused tensions and strains in health and welfare systems [3,4].

Most countries in the world have declared the chronological age of 65 as older age. Currently, there is no United Nations standard numerical criterion; however, the UN agreed that the cut-off age value for the older population is 60+ [5].

The Republic of Tatarstan (RT) is one of the most vibrantly developing regions of the Russian Federation with a population of 3.9 million people living in a territory of 67,836 square kilometres and has experienced, over the last three years, a 0.5% population growth. There has also been an increase in the ageing population, both in cities and the countryside, among men and women [4,6]. Therefore, the issues of healthy ageing are relevant for Russia and RT. Kazan is the capital of the Republic of Tatarstan, an ancient city with more than 1000 years of history, located on the left bank of the Volga River (https://www.kzn.ru/o-kazani/) with a current population of 1,257,391 people as of 1 January 2020 (the fifth location in Russia) [7,8]. Kazan is a vibrant city of sustainable economic growth and opportunities, comfortable for everyday life for its citizens, with a safe urban environment and age-friendly city [9].

Recently, an impressive number of initiatives, targeted at healthy lifestyle promotion, have been established and implemented in the Russian Federation and Tatarstan. However, there is a scarcity of published reports in the public domain. From our detailed searches of PubMed and eLIBRARY, the Russian database, we learnt that research on the ageing population, primarily of Moscow and Saint Petersburg, has been conducted since the 1990s using mostly the survey methodology [10,11]. The most extensive and recent monitoring of Moscow’s ageing population was performed in a framework of a longitudinal monitoring study over five years (2011–2015) by the centre for monitoring studies, with special emphasis on the indicators of the level and quality of life and attention to the changing needs of the ageing Muscovites [12]. Researchers have looked mostly at social and economic factors, emotions and quality of life of the older people. Over the last five years, Russian research attention has turned to information and training needs of older adults as a factor contributing to healthy ageing [13,14].

However, none of the published studies addressed or used the concept of an evidence-based approach to healthy ageing. We did not find any studies performed in Kazan, showing that there is an urgent need for raising evidence-based awareness amongst the ageing population of Kazan (Tatarstan, Russia). This might enable Kazan to join the European movement of transforming age-friendly cities to become age-inclusive cities [15].

Cochrane is an international organisation, a registered charity in the UK that has contributed to world health by pioneering and implementing the concept of organising medical research findings in a unique form of Cochrane systematic reviews. Cochrane systematic reviews facilitate informed decisions about treatment and other health interventions for better health. Over the last 25 years, Cochrane has gained international recognition for providing the benchmark of high-quality, unbiased and independent trusted evidence for better health [16,17]. Cochrane Russia was formed in Kazan 2015 to empower the Russian health system and Russian people with the best-synthesised research evidence.

From our practical work at Cochrane Russia, we recognise that the majority of current initiatives for healthy lifestyle promotion in Kazan and Tatarstan have been delivered by newly formed nongovernmental or private organisations utilising governmental funds run by young entrepreneurs who are unaware of research evidence, such as Cochrane, as witnessed by the results of our one-question survey at the meetings of health-promotion entrepreneurs. In our capacity of Cochrane Russia, we developed an Evidence School project for the ageing population of Tatarstan to improve health and longevity through Cochrane Knowledge Translation in the framework of the Cochrane consumers initiative Geographic Groups Consumer Engagement and Involvement Challenge Fund and were successful in our application. Here, we report the findings of the baseline survey.

Direct involvement of consumers in health care with shared decision making has evolved as a new approach or concept in health care over recent decades [18,19,20,21]. The language to define the new concepts and approaches in research and practice have evolved and require special attention to terminology that has not yet been fully unified [21]. The implementation of shared decision making should rely on knowledge and skills in evidence-based medicine, not only of health professionals but also of patients or consumers for quality and transparency of decision making [20]. Research in consumer training potential has emerged recently. It was shown that training in evidence-based medicine (EBM) was feasible [18,19] and better, and long-term implementation and further research are needed [18].

The aim of this study was to explore the health information needs of the ageing population of Kazan and the challenges people face in improving their health and longevity based on their attitudes, perceptions and assessment of their quality of life to ultimately raise evidence-based awareness about ageing. It is the most important action, mandated by the WHO, in order to ensure that societies become more just and fairer and institutions stronger to benefit the ageing population.

## 2. Materials and Methods

For this exploratory project, we performed a descriptive study among patients, caregivers and health care workers of the Interregional Clinical Diagnostic Centre (ICDC, Kazan, Republic of Tatarstan, Russian Federation), using the simple survey methodology. The Interregional Clinical Diagnostic Centre (ICDC) is one of the largest clinical hospitals of the Republic of Tatarstan, Kazan. It deals with the major burden of disease, disability and mortality of the Tatarstan population and cardiovascular diseases. The majority of regular ICDC patients belong to the older age groups. We planned to use the survey results as the basis for the Cochrane Evidence School development to enable targeting the most pertinent local health environment issues of the ageing population.

The survey consisted of 25 questions grouped into 3 sections: “Participant Portrait,” “Awareness Level” and “Exploring Ageing/Health Problems” and were a mixture of closed-ended and open-ended formats (the English translation of the survey questionnaire is presented in Appendix A).

The original plan was to focus on the ageing population only; however, at the time of the study, which coincided with the time of the expansion of the COVID-19 pandemic in Kazan, due to the scarcity of in-patients, we decided not to use any specific cut-off age value to include as many participants into our study as possible. All patients, caregivers and staff members of the ICDC, who agreed to participate in the study at that point of time, were included. This approach allowed us to determine whether there were any differences amongst various age groups in knowledge needs, perceptions, attitudes, quality of life, contentment with current position and challenges people face in maintaining and improving their health with ageing. Every participant gave their informed consent to participate in the study and anonymise the processing of personal demographic information (for an English translation of the Informed Consent Form, please see Appendix A and the Ethics Approval section) before taking part in the survey. They received a printed hard copy of the questionnaire at the ICDC and filled it in themselves or with the assistance of a caregiver over a period of 2 weeks in (10–25) April 2020.

Ethics Approval: the study obtained ethics approval from the Ethics Committee of the ICDC, numbered 259 dated 30th of March 2020. All the participants gave their informed consent before taking part in the survey. The English translation of the Informed Consent form is in Appendix A.

A total of 134 people participated in the study.

The survey results were combined in an Excel spreadsheet by the calculating absolute and relative numerical measurements for which we used the numbers of participants of various categories (age, gender, health profession, etc.) as the measurements. The results of open-ended questions were used in a descriptive way. Various subgroups of participants were compared to explore the potential associations of the studied parameters (answers to survey questions) with age, gender, representing health profession or not and other participant characteristics (but not a participant status with the ICDC (employee/patient/caregiver) using Fisher’s exact test [22]. The predefined significance threshold was *p* < 0.05 for a two-tailed test.

## 3. Results

Of a total of 134 people, who participated in the survey, there were: 44 patients of ICDC, 2 caregivers (family members) of the ICDC patients and 88 ICDC staff members.

### 3.1. General Characteristics of the Survey Participants, Their Lifestyle, Health Problems, Income, Quality of Life and Satisfaction

The distribution of participants by their general characteristics (gender, age, professional affiliation, etc.) is presented in Table 1. In total, 87 women (65%) and 47 men (35%) participated in the survey, and the majority of participants were under 60 years old (117/134; 87%). More than half of the participants had higher or secondary specialised medical/pharmaceutical education (73/134; 54%) and were healthcare workers (86/134; 64%).

Most participants (115/134; 86%) indicated that they had various chronic diseases or long-term health conditions, and many of them had more than one health problem. Among the most frequently mentioned were cardiovascular diseases (37/134; 28%), gastrointestinal diseases (31/134; 23%), bone and joint diseases (25/134; 19%), neurological diseases (22/134; 16%) and respiratory diseases (13/134; 10%).

The majority of participants considered their lifestyle to be unhealthy (92/134; 69%), including having a sedentary lifestyle (33/134; 25%), a lack of attention to their health (28/134; 21%), an unhealthy diet (26/134; 19%), smoking (21/134; 16%) and excess alcohol consumption (3/134; 2%).

The income level of the participants ranged from under 8 thousand rubles per month to over 90 thousand rubles per month, and the median income level among all participants was 20–30 thousand rubles per month. For the majority of participants (>80%), the income level ranged from 12 to 60 thousand rubles. The income level of participants over 60 years old (16 people) also ranged from under 7–8 thousand rubles to over 90 thousand rubles per month, and almost half of the participants (7 out 16 people) had an income level within 12–20 thousand rubles.

Regarding the frequency of medicine use, 52 out 134 people (38%) responded that they rarely took medicines, 39 people (29%) took medicines every day, 29 people (22%) took medicines periodically, 13 people (10%) answered never and 1 person did not answer this question. However, when comparing the frequency of taking medicines in the age groups of our interest (60+ vs. <60), it was found that more people aged 60+ (10 out 16 people) took medicines every day whereas more people under 60 years of age (49 out 132 people) took medicines rarely (Table 2).

When comparing the answer “daily use” with all other answers combined among people aged under 60 and aged 60+, it was found that more of the older participants took medicines on a daily basis: 10 out of 16 compared to 29 out of 117 of people under 60 (*p* = 0.006, Fisher’s exact test).

There were also differences in medicine use between men and women: half of the male participants answered that they took medicines every day whereas fewer than one-fifth of women responded that they took medicines every day (Table 3). Most of the female participants responded that they took medicines rarely (38 out 87 women; 44%) or periodically (25 out 87 women; 29%).

When comparing the answer “daily use” with all other answers combined among men and women, it was found that more of the men than women took medicines on a daily basis (*p* = 0.0003, Fisher’s exact test).

Thus, medicine use was heavier in the older participants (aged 60+) and men.

All participants were asked to rate their quality of life on a 10-point scale. We attributed a score of 1–4 points to low quality of life, 5–7 points to moderate quality of life and 8–10 points to high quality of life.

Out of 134 participants, 131 assessed the quality of their life, but since 1 of them did not indicate her age (woman with low quality of life), we were able to compare the quality of life among 130 participants with a known age. The results of assessing the quality of life of survey participants by age (under 60 years vs. 60+ years) are presented in Table 4.

Many participants rated their quality of life high enough: more than half of the participants (69/133; 52%) considered their quality of life to be high, and 48 out 133 participants (36%) considered their quality of life to be moderate.

We combined participants with high and moderate quality of life together as participants with satisfactory quality of life and compared the quality of life (satisfactory vs. unsatisfactory) among participants aged under 60 years and 60+ years (Table 5). Low quality of life was considered unsatisfactory. Four participants were excluded from this assessment because three of them did not assess their quality of life and one did not indicate her age.

The proportion of the participants with unsatisfactory quality of life was higher among participants aged 60+ compared to the participants aged under 60. However, the number of participants over 60 was small. With the use of Fisher’s exact test to assess the significance of differences between compared groups, the probability of such a result was found to be approximately 0.03. This means that we can refute the null hypothesis that there are no differences between groups, and the quality of life of participants aged 60+ is worse compared to participants aged under 60.

It was important to determine if the quality of life of the participants changed with age in their opinion and, if changed, what the changes were. For the question “Has the quality of life changed with age?” 52 out of 134 people (39%) answered “yes” (9 people aged 60+, 42 people aged under 60 and 1 person with unknown age). Twelve out of these 52 people answered that their quality of life improved with age (1 person aged 60+ and 11 people aged under 60). However, 22 out of 52 people answered that their quality of life worsened with age (4 people aged 60+ and 18 people aged under 60).

The analysis of the responses regarding participants’ satisfaction with their current situation and their life in general showed that more than half of the participants (73/134; 55%) answered that they were satisfied with their current situation, 39 out 134 participants (29%) answered that they were unsatisfied, 21 out 134 participants (16%) answered that they had difficulties with this question and 1 participant did not answer at all.

When comparing participants’ satisfaction with their current situation between the two age groups (60+ vs. under 60), there were no significant differences after using Fisher’s exact test (Table 6). Two participants were excluded from this assessment because one of them did not assess her quality of life (woman >80 years) and one participant (unsatisfied) did not indicate her age. Thus, slightly more than half of the participants, regardless of age, were satisfied with their current situation.

The participants who were unsatisfied with their current situation were asked to answer what they would like to change. Twenty-two people out of 38 answered. Many participants responded they would welcome positive changes in their wellbeing, health and lifestyle (nine people), in their financial situation (nine people) and some of them would like changes in professional development (career), health care, family support and personal life.

### 3.2. Age-Related Problems, Awareness about Ageism and Participation in Antiaging Programmes

Since at the inception of the study it was planned to include mostly older people to participate in our survey, one of the questions of our questionnaire asked if people experienced age-related problems. Thirty-one out 134 participants (23%) answered that they had encountered problems related to their age (21 people aged under 60, 9 people aged 60+ and 1 person with unknown age).

When comparing the proportion of participants who had encountered age-related problems or not within the two age groups (60+ vs. under 60), it was found that older participants (60+) reported age-related problems more frequently (Table 7).

When using Fisher’s exact test to assess the association of reporting on age-related problems with age, it was found that the probability of the association was there with the value of approximately *p* = 0.002. Unsurprisingly, the participants aged 60+ encountered more age-related problems.

However, one cannot be sure that the participants reported about their personal experiences of age-related problems, particularly those of the younger participants. It cannot be ruled out that they meant not only their personal experience but also the experience of their relatives or friends.

The majority of participants (17 out 30 people; 12 people aged from under 30 to 60, 4 people aged 60+ and 1 person with unknown age) described these problems as health-related problems (diseases, impairment of vision and hearing, memory and endurance, as well as overweight, decreased alertness and reaction, and fatigue). Some participants (3 people aged 50–70) reported social problems (deprivation of work, decrease in income (salary), lack of funds for living) and one participant (aged 40–50) noted problems associated with appearance (wrinkles).

It was important to know if our participants had heard about ageism and, if yes, what it was in their opinion. According to Wikipedia, ageism is determined as “stereotyping and/or discrimination against individuals or groups on the basis of their age” [23]. According to the WHO, ageism is defined as “*stereotypes, prejudice, and discrimination against older people on the basis of their chronological or perceived age.*” it is widespread and has harmful effects on the health of older adults, their social life, employment and other aspects of life [1].

It was found that only 13 out of 134 participants (10%; 7 men and 6 women) previously heard about ageism (Table 8), including 11 people aged under 60 and 2 people aged 60+. Nearly all of them were health care providers (12 people). There were no differences in the awareness about ageism between the two age groups (60+ vs. under 60).

Only nine participants were able to correctly explain what ageism was. It could be suggested that the low awareness about ageism among our survey participants may be due to insufficient use of this English-language-originating term in Russia and the absence of a Russian-language analogue of this term. Since the number of participants was small, further research with more people is needed to study the awareness of the phenomenon of ageism.

Regarding participation in antiaging programmes, only 2 out of 134 people answered that they had previously participated in antiaging programmes. Two men aged 40–50 and 60+, both health care providers, with an income level of 60–90 thousand rubles per month, who rated their quality of life as high and were satisfied with their current situation, had never heard about ageism before. One of them thought that participation in antiageing programmes was beneficial for him to facilitate quitting bad habits, and the other considered it to aid in improving his wellbeing.

It could be assumed that not all of the participants understood what antiaging programmes were since there was no explanation in the question and it was the English term “antiaging” that was used in the questionnaire (Appendix A). Perhaps additional clarifications to this question and the use of the Russian-language analogue of the name of antiaging programmes would lead to different results of the answer to this question.

### 3.3. Awareness about Evidence-Based Medicine, Cochrane and Health Information Needs

The specially designed series of questions asked about awareness of evidence-based medicine (EBM), Cochrane and the needs for health information or knowledge (willingness to participate in the Evidence School).

Fifty-five out 134 participants (41%) had previously heard about evidence-based medicine (50 people aged 60+ and 4 people aged under 60; 25 men and 30 women; 42 health care providers, 9 non-health-care providers and 4 people retired (unemployed)). There were no significant differences in awareness about evidence-based medicine between the two age groups (60+ vs. under 60) after using Fisher’s exact test. Only a weak tendency to the potential association that awareness was higher in participants aged under 60 (*p* = 0.2) could be suggested. It was noteworthy that EBM-awareness was higher among men than women (*p* = 0.04).

Assuming that EBM-awareness should be higher among health care providers, we compared the proportion of participants who had previously heard about EBM among health care providers and non-health-care workers, combined with retired (and unemployed) participants (Table 9).

EBM-awareness was higher among health care providers compared to non-health-care workers combined with nonworking participants (Fisher’s exact test, *p* = 0.01).

The survey asked about participants’ awareness of Cochrane and Cochrane evidence and contribution as Cochrane is the leading international organisation for evidence synthesis, knowledge translation, and independent quality research advocacy.

It was found that the majority of participants (93 out of 134 people; 70%) had not heard about Cochrane. Out of 134 people, 34 (25%) answered that they were aware of Cochrane and Cochrane work, and 7 out 134 people (5%) could not answer this question.

When comparing Cochrane awareness between the two age groups of the participants (60+ vs. under 60), no differences were found.

When comparing the numbers of aware and unaware (in relation to Cochrane) participants between health care providers and non-health-care workers, it was found that Cochrane awareness was associated with health care professionals both when compared to unaware people (*p* = 0.001) and when compared to the combined group of unaware and people experiencing difficulties to answer (*p* = 0.000008) (Table 10).

Thus, our survey confirmed that awareness about evidence-based medicine and Cochrane was higher within the healthcare profession than all others. However, it turned out that more than one-third of health care providers had never heard about EBM (31 out 86 health care workers; 36%) and more than half had never heard about Cochrane (50 out 86 health care providers; 58%).

In the framework of our project of Evidence School for the ageing population of Tatarstan aimed at improving health and longevity, it was planned to explore the needs in the knowledge of EBM and Cochrane for further designing the Evidence School project. Forty-six out of 134 people answered that they were interested in learning (34%), 34 people (25%) answered that they were not interested, 44 people (33%) could not answer and 10 people (8%) did not answer.

When comparing the need for knowledge about EBM and Cochrane between the two age groups (60+ vs. under 60), as well as between men and women, participants with satisfactory or unsatisfactory quality of life, no significant differences were found. Initially, assuming that the knowledge needs are possibly higher in health care workers compared to other workers or nonworking people, it was important to check if there are differences in knowledge needs between them (Table 11).

The need for knowledge about EBM and Cochrane was higher among health care providers in the following comparisons:(1)“Interested in gaining knowledge (learning)” vs. “not interested” amongst health care providers and non-health-care workers (*p* = 0.003);(2)“Interested in gaining knowledge (learning)” combined with “difficult to answer” vs. “not interested” amongst health care providers and non-health-care workers (*p* = 0.009);(3)“Interested in gaining knowledge (learning)” vs. “not interested” amongst health care providers and non-health-care workers combined with retired (not working) participants (*p* = 0.001);(4)“Interested in gaining knowledge (learning)” combined with “difficult to answer” vs. “not interested” amongst health care providers and non-health-care workers combined retired (not working) participants (*p* = 0.01).

## 4. Discussion

A descriptive study (survey) was performed among patients, caregivers and health care providers of the Interregional Clinical Diagnostic Centre (Kazan, Republic of Tatarstan, Russian Federation), using the simple survey methodology for this exploratory project. The age of the participants ranged from under 30 to over 80 years. The majority of the participants were people under 60 years old, women, healthcare providers with medical or pharmaceutical education, with long-term health conditions and an unhealthy lifestyle. This age disposition of participants is not representative of the current population of Tatarstan because, in 2016, Tatarstan moved to the very high level of demographic old age, with people aged 60+ comprising 18% of the population and more [4]. However, it is characteristic of the pandemic situation at the ICDC and loosely reflects the age disposition of in-patients in Russia [4,10]. The majority of the participants rarely took medicines, but there were more participants taking medicines more often (daily) among people aged 60+ and men, which is, again, in line with the findings of Belokon on the importance of medicines and reliance on them within the retired Russian population [10].

Most participants positively assessed their quality of life: half noted high quality of life, and over one-third of participants noted moderate quality of life. Ageing contributed to the lower perception of quality of life. The proportion of participants who rated their quality of life as unsatisfactory was higher among participants aged 60+. This is in line with the findings of the large-scale (about 700 respondents) survey in six territories of Russia in the vicinity of Moscow, reporting low quality of life of older adults. The survey showed that the low quality of life was due to low (below the subsistence level) pensions, but was not limited to this financial aspect only, demonstrating that the older people perceived their quality of life low because of the lack of respect for their personality [10].

Most participants in our study noted that their quality of life has changed with age negatively primarily due to health problems. Very few participants included financial problems to their characteristics of poor quality of life; however, some participants noted a positive change in their quality of life with age, describing more emerging opportunities, freedom and becoming smarter. More than half of the participants were satisfied with their current situation without any age differences. These findings suggest that, indeed, the ageing population of Kazan may be in a better situation compared to the rest of the country. Though this assumption needs to be confirmed in larger-scale studies, it could be suggested that Kazan is closer to age-friendly cities than others in the country.

It could be assumed that the significance of the different components of quality of life varies among different people, and this is probably a rather subjective assessment. Although some participants who reported an improvement in the quality of their life with age noted that they became smarter with more opportunities and more freedom, the participants who reported worsening of their quality of life with age noted an increase in health problems. Thus, the participants assessed the quality of life from different perspectives, both from social situations, interactions and health conditions, depending on what affected and worried them most. This is in line with the findings of a large community survey of retired citizens of Saint Petersburg (1500 people), which identified social components of the quality of life of older adults with mainly the worsening of financial status and becoming less satisfied with their current situation and more reliant on their children in taking responsibility for their wellbeing [11]. Certainly, it could be assumed that the age of participants in our study might have affected their perception of the change in quality of life with age, though we could not assess this factor quantitatively. For instance, participants aged 60+ more often encountered age-related problems, primarily perceiving them as health-related problems, commented less often on financial or work-related problems. This perception could be one of the potential reasons for the low awareness of ageism among our participants. However, there may be simply a lack of familiarity with the term per se. Noteworthy is the fact that none of the identified publications on ageing problems in Russia used this term, though studied components of what constitutes ageism [10].

The awareness of evidence-based medicine and its approaches was low. Less than half of participants were aware of EBM and only a quarter were aware of Cochrane. Health professionals were more informed about EBM and Cochrane, and age did not contribute to better or poorer awareness. This is our pioneering finding, and though it needs verification and confirmation in larger studies, we will use the data from this survey to design and conduct the Cochrane Evidence School at the ICDC. This will actively involve both patients and healthcare providers in the design and production of Cochrane Knowledge Translation products for their personal use and use by their peers. We will survey participants of the School at its first-round completion and report or new findings. We think that this work will help us to plan further research in this area and design a more informed plan of action for better Cochrane Knowledge Translation specifically tailored to the needs of the ageing population of Kazan.

We hope to bring closer the times of Kazan becoming not only a smart but truly age-friendly [24] and even an age-involved city when society will get to the position of being able to celebrate the current increases in longevity and life expectancy. Our project will empower the ageing population of Kazan and further raise awareness of the need for evidence and “action to ensure good health in older ages” following the Cochrane Campbell Global Ageing Partnership and its Wikipedia Project [25]. We hope that our findings and future development of the project will contribute to the expanding area of research on the proposed extension of the WHO’s age-friendly cities model by following its recommendations on including citizens from all sectors of society, consultations and data collection, including a qualitative approach to measure attitudes and perceptions [2].

Finally, about a third of the surveyed people were interested in new knowledge and learning (participating in Cochrane Evidence School). Unsurprisingly, the need for knowledge was higher among healthcare providers, but otherwise, there were no differences between participants aged 60+ or under 60, men and women, participants with satisfactory or unsatisfactory quality of life and other characteristics. This is in line with the findings of the survey study in Saratov Day-Care Centre for Social Services the Population of Saratov (for retired people), which looked at the mechanisms and drivers of social activity of older citizens [14]. The author identified internal resources of the older people for voluntary work, namely spare time, experience, knowledge, the desire to contribute and components of motivation: “the need for communication and the desire to be socially useful to other people; the need for new knowledge and, accordingly, in new social roles. Retirement leisure in the social service forms an active position (physical, educational activity). The results of the research showed the presence of interest and the desire to take the initiative in organising social events” [14].

These findings together with the results of our exploratory study provide a good background for the implementation of Cochrane Evidence School and later hopefully scaling up the experience and the impact to contribute to the vision for an age-friendly ecosystem and prevention-focused public health system [26].

### Strength and Limitations

This study is the first to explore the concept of an evidence-based approach and specifically Cochrane for healthy ageing and to address a number of factors contributing to the challenges of ageing in Kazan, Tatarstan, Russia. Data were collected with the same standardised questionnaire in all participants and analysed the data quantitatively and qualitatively.

When planning the study, we aimed to survey 50 people but did not limit the target number in case more people volunteered to participate, ending up with nearly three times the initial target.

All patients of the ICDC (at the time of the survey (end of March–beginning of April 2020), Russia was under a strict lockdown period, and only emergency patients were at the ICDC) were included as well as their family members who agreed to participate. This allowed us to subgroup participants by age to determine if there were any differences amongst various age groups in knowledge, perceptions, attitudes and challenges people face in maintaining and improving their health with ageing. Involving health care providers allowed for a new dimension of the study and comparison of answers from consumers to the answers from within the health profession.

However, the findings of this exploratory study should be interpreted with caution and need to be confirmed in a larger-scale research project owing to the number of limitations: small numbers of survey participants from a single health care setting, small and unequal numbers of participants in compared subgroups; substantial representation of health professionals and people under 60 years of age. The possibility that some of the questions were not clear enough to all the participants could not be ruled out, although there is hope that participants answered the questions with full sincerity.

Although the sample is small, despite having over 130 participants, this is the first exploratory study in a region of Russia that has a diverse community (Tatar/Russian), and this work forms a basis to progress this domain in Russia across rural and urban environments.

## 5. Conclusions

This exploratory study provides data on the health information needs of the ageing and middle-aged population of Kazan by documenting that less than half of the survey participants were aware of evidence-based medicine and only a quarter of them were aware of Cochrane and Cochrane work. EBM-awareness was higher within the healthcare profession than others and among men than women, with a weak tendency to be higher in participants under 60 years of age than in people of 60+ years old. Cochrane awareness was also higher within the healthcare profession. The need for EBM and Cochrane knowledge was also higher within the health profession without age or gender differences or between participants with satisfactory and unsatisfactory quality of life.

Most of the survey participants were generally positive in assessing their quality of life despite the fact that many have long-term health conditions and unhealthy lifestyles. A few reported improvements in quality of life with age.

Older participants were less positive about their quality of life. More participants aged 60+ took medicines on a daily basis than people under 60 years of age.

Ageing was the challenge in improving health and longevity with more participants aged 60+ rating their quality of life as unsatisfactory than participants under 60 years old, and encountering more age-related problems. Only 10% of participants were aware of ageism without age or gender differences.

Low awareness of EBM, Cochrane, ageing problems and ageism, together with the scarcity of independent, accessible, quality information about the effectiveness and safety of therapeutic and diagnostic interventions, nondrug treatment and prevention measures, present a case for the urgent need for raising awareness of ageing problems, ageism and evidence-based approach to tackling the problems of the ageing population of Kazan, which strives to be an age-friendly city.

Cochrane Evidence Schools designed not only for patients and consumers but also for health care providers might become a beneficial instrument both for raising awareness and advancing knowledge and attitudes of the citizens of Kazan as a pilot city with the potential of scaling up.

The hope is that this exploratory study would contribute to Kazan becoming not only a fully age-friendly city but moving towards age-inclusiveness.

## Figures and Tables

**Table 1 ijerph-17-09212-t001:** General characteristics of the survey participants.

**Age**	***N* (%)**	**Gender**	***N* (%)**
Under 60 years	117 (87%)	Women	87 (65%)
60+ years	16 (12%)	Men	47 (35%)
Unknown	1 (1%)		
Total (%)	134 (100%)	Total (%)	134 (100%)
**Education**	***N*** **(%)**	**Occupation**	***N*** **(%)**
Higher or secondary specialised medical/pharmaceutical education	73 (54%)	Health care providers	86 (64%)
Higher/secondary specialised education (excl. med/pharm)	53 (40%)	Non-health-care workers	23 (17%)
Secondary education	8 (6%)	Retired	25 (19%)
Total (%)	134 (100%)	Total (%)	134 (100%)
**Unhealthy Lifestyle (Including Smoking, Alcohol Consumption, Etc.)**	***N*** **(%)**	**Chronic Diseases (Long-Term Health Conditions)**	***N*** **(%)**
Yes	92 (69%)	Yes	115 (86%)
No	19 (14%)	No	19 (14%)
Unknown (no answer)	23 (17%)		
Total (%)	134 (100%)	Total (%)	134 (100%)
**Income Level Per Month**			
Under 8000 RUB (88 Euro)			3 (2%)
8000–12,000 RUB (88–133 Euro)			9 (7%)
12,000–20,000 RUB (133–221 Euro)			29 (21%)
20,000–30,000 RUB (221–332 Euro)			39 (29%)
30,000–60,000 RUB (332–663 Euro)			43 (32%)
60,000–90,000 RUB (663–995 Euro)			8 (6%)
>90,000 RUB (995 Euro)			2 (2%)
Unknown			1 (1%)
Total (%)			134 (100%)

*N*–number, %–percentage of total number.

**Table 2 ijerph-17-09212-t002:** Frequency of medicine use among the participants by age.

Frequency of Medicine Use	Number of Participants Aged Under 60	Number of Participants Aged 60+	Total Number of Participants
Daily	29 *	10 *	39 (29%)
Periodically	26	3	29 (22%)
Rarely	49	2	51 (38%)
Never	12	1	13 (10%)
No answer	1		1 (1%)
In total	117	16	133 (100%)

* *p* = 0.006 (Fisher’s exact test).

**Table 3 ijerph-17-09212-t003:** Frequency of medicine use among the participants by gender.

Frequency of Medicine Use	Number of Men	Number of Women	Total Number of Participants
Daily	23 *	16 *	39 (29%)
Periodically	4	25	29 (22%)
Rarely	14	38	52 (38%)
Never	5	8	13 (10%)
No answer	1		1 (1%)
In total	47	87	134 (100%)

* *p* = 0.0003 (Fisher’s exact test).

**Table 4 ijerph-17-09212-t004:** Participants’ quality of life (self-reported) by age.

Participants’ Quality of Life	Number of Participants Aged under 60	Number of Participants Aged 60+	Total Number of Participants
High (8–10 points)	65	4	69 (52%)
Moderate (5–7 points)	42	6	48 (36%)
Low (1–4 points)	9	4	13 (10%)
Unknown	1	2	3 (2%)
Total	117	16	133 (100%)

**Table 5 ijerph-17-09212-t005:** Satisfactory and unsatisfactory quality of life (self-reported) by age.

Participants’ Quality of Life	Number of Participants Aged under 60	Number of Participants Aged 60+	Total Number of Participants
Satisfactory (5–10 points)	107 *	10 *	117 (90%)
Unsatisfactory (1–4 points)	9 *	4 *	13 (10%)
Total	116	14	130 (100%)

* *p* = 0.03 (Fisher’s exact test).

**Table 6 ijerph-17-09212-t006:** Participants’ satisfaction with their current situation.

Participants’ Satisfaction	Number of Participants Aged under 60	Number of Participants Aged 60+	Total Number of Participants
Satisfied	65	8	73 (55%)
Unsatisfied	32	6	38 (29%)
Difficult to answer	20	1	21 (16%)
Total	117	15	132 (100%)

**Table 7 ijerph-17-09212-t007:** Experience of age-related problems by age.

Have You Encountered Any Problems Related to Your Age?	Number of Participants Aged under 60	Number of Participants Aged 60+	Total Number of Participants
Encountered	21 *	9 *	30 (22%)
Not encountered	84	6	90 (68%)
Difficult to answer	12	0	12 (9%)
No answer	0	1	1 (1%)
Total	117	16	133 (100%)

* *p* = 0.002 (Fisher’s exact test).

**Table 8 ijerph-17-09212-t008:** Participants’ awareness of ageism.

Awareness about Ageism	Number of Participants Aged under 60	Number of Participants Aged 60+	Total Number of Participants
Aware (previously heard) about ageism	11	2	13 (10%)
Unaware (not previously heard) about ageism	60	14	74 (55%)
Difficult to answer	5	0	5 (4%)
No answer	41	0	41 (31%)
Total	117	16	133 (100%)

**Table 9 ijerph-17-09212-t009:** Participants’ awareness of evidence-based medicine (EBM).

Awareness about (EBM)	Number of Health Care Providers	Number of Non-Health-Care Workers	Number of Retired Participants	Total Number of Participants
Aware about EBM	42 *	9 *	4 *	55 (41%)
Unaware about EBM	31	4	13	48 (36%)
Difficult to answer	13	9	9	31 (23%)
Total	86	22	26	134 (100%)

* *p* = 0.01 (Fisher’s exact test).

**Table 10 ijerph-17-09212-t010:** Participants’ awareness of Cochrane and Cochrane work.

Awareness about Cochrane	Number of Health Care Providers	Number of Non-Health-Care Workers	Number of Retired Participants	Total Number of Participants
Aware of Cochrane	32	1	1	34 (25%)
Unaware of Cochrane	50	20	23	93 (70%)
Difficult to answer	4	2	1	7 (5%)
Total	86	23	25	134 (100%)

**Table 11 ijerph-17-09212-t011:** Need for knowledge about evidence-based medicine (EBM) and Cochrane.

Need for Knowledge about EBM And Cochrane	Number of Health Care Providers	Number of Non-Health-Care Workers	Number of Retired Participants	Total Number of Participants
Interested in gaining knowledge (learning)	38	3	5	46 (34%)
Not interested in gaining knowledge (learning)	16	10	8	34 (25%)
Difficult to answer	27	7	10	44 (33%)
No answer	5	3	2	10 (8%)
Total	86	23	25	134 (100%)

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
