# Peer review of "Smart and Age-Friendly Cities in Russia: An Exploratory Study of Attitudes, Perceptions, Quality of Life and Health Information Needs"

_ijerph, 2020, doi:10.3390/ijerph17249212_

Round 1

Reviewer 1 Report

General comments

This study collected age-friendliness in Kazan, Russia. The questionnaire did not collect adequate information to achieve the authors’ stated aims. The authors only collect data about whether the participants “heard of evidence-based medicine”, “heard of Cochrane”, “heard of ageism”, how did they obtain health information, and a 1-10 score on their quality of life. I don’t think there are any useful information collected in this survey with 134 participants.

Specific comments

Line 23. If this is a pilot study, then when and how will the main study be conducted?

Lines 28-32. Please quantify the associations here.

Line 104. A lot of information was missed here. How were the participants recruited? How were participant consent obtain? Which institution approved this study? The measurement details (lines 110-112) were inadequate.

Lines 123-124. The details of how the data / variables were processed should be provided.

Lines 128-131. It belongs to Methods.

Line 132. If there were only 2 caregivers, they should not be included in the data analysis.

Line 142. The characteristics of the participants should be stratified by patients and staff. The variable “unhealthy lifestyle” is too broad.

Lines 164, 173, 185, 193, 218, 232, 256, 288, 304, and 319. Column percentage and the results of Fisher exact test should be provided.

Author Response

Cover / rebuttal letter

Dear editors and reviewers,

With this letter we thank you for the detailed peer-review process and helpful suggestions for improvement of our manuscript “Smart and Age-Friendly Cities in Russia: A pilot survey on attitudes, perceptions, quality of life and health information needs” by Liliya Eugenevna Ziganshina, Ekaterina V. Yudina, Liliya I. Talipova, Guzel N. Sharafutdina, and Rustem N. Khairullin.

We are uploading the revised version to the Editorial manager.

Our answers to reviewers’ comments and suggestions:

Reviewer 1

General comments

This study collected age-friendliness in Kazan, Russia. The questionnaire did not collect adequate information to achieve the authors’ stated aims. The authors only collect data about whether the participants “heard of evidence-based medicine”, “heard of Cochrane”, “heard of ageism”, how did they obtain health information, and a 1-10 score on their quality of life. I don’t think there are any useful information collected in this survey with 134 participants.

Authors’ answer:

Thank you very much for the general comments. We understand your concerns and would like to say here that we think that in the framework of the pilot (first ever in Russia in this field at the regional level) study, the survey results are indeed useful. They are useful for:

  • background assessment of the needs and problems of aging population,
  • selection of participants for Evidence School,
  • planning of training (based on the information received) and
  • conducting a follow-up study (survey) after training, as well as
  • planning for larger studies (scaling up), taking into account the results obtained and identified shortcomings, which would allow for improvement of the methodology, including the questionnaire.

Understanding the background level of awareness / knowledge prior to our Cochrane intervention was the main objective of our pilot study: The aim of the study was to explore the health information needs of ageing population of Kazan and the challenges people face in improving their health and longevity based on their attitudes, perceptions and assessment of their quality of life in Kazan to ultimately raise evidence-based awareness about ageing.

We also included one more paragraph in the Introduction section to make our stated aims better substantiated:

“Direct involvement of consumers in health care with shared decision making has evolved as a new approach or concept in health care over the last decades [17, 18, 19, 10]. The language to define the new concepts and approaches in research and practice evolved and required special attention to terminology which has not yet been fully unified [20]. Implementation of shared decision making should rely on knowledge and skills in evidence-based medicine not only of health professionals but also of patients or consumers for quality and transparency of decision making [19]. Research in consumer training potential emerged recently. It was shown that training in EBM was feasible [17, 18], better and long-term implementation and further research are needed [17].”

Specific comments

Line 23. If this is a pilot study, then when and how will the main study be conducted?

Authors’ answer:

Thank you very much for his comment. Indeed, we indicated in the abstract that this is a pilot study (lines 23-24 of the original and revised manuscript). We present the outline of the entire project in the section 2. Materials and Methods (lines 104-126 of the original manuscript, where we also showed the time of the initial stage of the project), and further emphasize the pilot nature of the study in the subsection Strength and Limitations (lines 425-444 of the original manuscript; lines 498-517 of the revised manuscript), as well as throughout the manuscript, while the word count limit does not allow for such detailing. In the manuscript we do not provide any definitive dates of the next stages of the entire project and will not be able to indicate any definitive dates now in this answer to the comments because the changing pandemic situation moved our initial plans and we do not know yet how it will evolve. We will proceed to the planned next stages as soon as it will become possible.            

Lines 28-32. Please quantify the associations here.

Authors’ answer:

Thank you very much for this comment. We quantified the associations, introducing the numbers into the corresponding lines of the abstract (28 to 35) to read:

“Older people (60+) were less positive about their quality of life, more often took medicines on daily basis (10/16 compared to 29/117 of people under 60), encountered problems with ageing (9/16 compared to 21/117 of people under 60) and rated their quality of life as unsatisfactory (4/14 compared to 9/107 of people under 60). Awareness in EB approaches and Cochrane was higher within health profession (evidence-based medicine: 42/86 vs 13/48; Cochrane: 32/86 vs 2/48), health information needs did not differ between age or gender groups or people with satisfactory and unsatisfactory quality of life. The minority (10% - 13/134) were aware of ageism without age or gender differences.”

Line 104. A lot of information was missed here. How were the participants recruited? How were participant consent obtain? Which institution approved this study? The measurement details (lines 110-112) were inadequate.

Authors’ answer:

Thank you very much for this comment. We modified the description of how participants were recruited, included description of anonymity / confidentiality and the referral to the Ethics approval, ICDC (institution), the Informed Consent Form and the measurements to read (lines 126 to 145 of the revised manuscript):

“The original plan was to focus on ageing population only, however, at the time of the pilot, which coincided with the time of expansion of COVID-19 pandemic in Kazan, due to the scarcity of in-patients we decided not to use any specific cut-off age value to include participants into our pilot. All patients, caregivers and staff members of the ICDC, who agreed to participate in the pilot at that point of time, were included. This approach allowed us to see whether there were any differences amongst various age groups in knowledge needs, perceptions, attitudes, quality of life, contentment with current position and challenges people face in maintaining and improving their health with ageing. Every participant gave his/her informed consent to participate in the pilot and to anonymized processing of personal demographic information (for the English translation of the Informed Consent form please see Appendix S2 and the Ethics approval section) before taking part in the survey and received a printed hard copy of the questionnaire at the ICDC and filled it in by him or herself or a care-giver over the period of 2 weeks of April (10-25) 2020. A total of 134 people took part in the study.

The survey results were combined in Excel spreadsheet by calculating absolute and relative numerical measurements for which we used numbers of participants of various categories (age, gender, health profession, etc) as the measurements. The results of open-ended questions were used in descriptive way. Various subgroups of participants were compared with each other to explore potential associations of studied parameters (answers to survey questions) with age, gender, representing health profession or not and other participant characteristics (but not a participant status with the ICDC (employee / patient / caregiver) using Fisher’s exact test.”

Here is the text of the Ethics approval subsection (lines 559 to 561 of the revised manuscript):

“Ethics approval: the study obtained ethics approval from the Ethics Committee of the ICDC, numbered 259 dated 30th of March 2020. All the participants gave their informed consent before taking part in the survey. The English translation of the Informed Consent form is in Appendix S2.”

Lines 123-124. The details of how the data / variables were processed should be provided.

Authors’ answer:

Thank you very much for this comment. We modified the description of data procession to read (lines 139 to 145 of the revised manuscript):

”The survey results were combined in Excel spreadsheet by calculating absolute and relative numerical measurements for which we used numbers of participants of various categories (age, gender, health profession, etc) as the measurements. The results of open-ended questions were used in descriptive way. Various subgroups of participants were compared with each other to explore potential associations of studied parameters (answers to survey questions) with age, gender, representing health profession or not and other participant characteristics (but not a participant status with the ICDC (employee / patient / caregiver) using Fisher’s exact test.”

Lines 128-131. It belongs to Methods.

Authors’ answer:

Thank you very much for this comment. We moved the description of the ICDC to the Methods section to read (lines 114 to 121 of the revised manuscript):

“We performed a descriptive study among patients, caregivers and health care workers of the Interregional Clinical Diagnostic Centre (ICDC, Kazan, Republic of Tatarstan, Russian Federation), using simple survey methodology for this pilot project. The Interregional Clinical Diagnostic Centre (ICDC) is one of the largest clinical hospitals of the Republic of Tatarstan, Kazan. It deals with the major burden of disease, disability and mortality of Tatarstan population, cardiovascular diseases. The majority of regular ICDC patients belong to the older age groups. We planned to use the survey results as the basis for Cochrane Evidence School development to enable targeting the most pertinent to local health environment issues of ageing population.”

Line 132. If there were only 2 caregivers, they should not be included in the data analysis.

Authors’ answer:

Thank you very much for this comment. Since we did not compare staff with patients and with caregivers, we did not exclude caregivers from the data analysis. In these lines we presented these data to describe participants in general, as a characteristic of the participants in relation to the ICDC (staff, patients, caregivers). But since some of the staff were of non-medical profession, while some of the patients were of medical profession, we decided that it would be important to compare the answers of people representing health profession versus the answers of people not belonging to health profession, and not depending on the status at the ICDC (employee / patient / caregiver). In other words we did not single out those two caregivers in our analyses. To make this explicitly clear we provided the details in the methods section, which also answer one the above comment (lines 139 to 145 of the revised manuscript) to read:

”The survey results were combined in Excel spreadsheet by calculating absolute and relative numerical measurements for which we used numbers of participants of various categories (age, gender, health profession, etc) as the measurements. The results of open-ended questions were used in descriptive way. Various subgroups of participants were compared with each other to explore potential associations of studied parameters (answers to survey questions) with age, gender, representing health profession or not and other participant characteristics (but not a participant status with the ICDC (employee / patient / caregiver) using Fisher’s exact test.”

Line 142. The characteristics of the participants should be stratified by patients and staff. The variableunhealthy lifestyleis too broad.

Authors’ answer:

Thank you very much for this comment. Since we did not compare staff with patients and with caregivers, we did not exclude caregivers from the data analysis. In these lines we presented these data to describe participants as a whole, as a characteristic of the participants in relation to the ICDC (staff, patients, caregivers). But since some of the staff were of non-medical profession, while some of the patients were of medical profession, we decided that it would be important to compare the answers of people representing health profession versus the answers of people not belonging to health profession, and not depending on the status at the ICDC (employee / patient / caregiver), which could introduce bias. We agree that the variable “unhealthy lifestyle” or “chronic diseases” are rather broad. However, we think that operationalising by Yes/No (having any features of unhealthy lifestyle or not; and having a chronic disease or chronic diseases or not) allowed for accurate participant categorisation, particularly given the small number of participants in our pilot study. 

Lines 164, 173, 185, 193, 218, 232, 256, 288, 304, and 319. Column percentage and the results of Fisher exact test should be provided.

Authors’ answer:

Thank you very much for this comment. We provide the percentages in brackets and the Fisher’s exact test results for every comparison in the text in full detail. The reason we could not include the Fisher’s exact test results in a separate column of our tables is that in the tables we provide numerical variables (numbers of participants by certain categories following various survey answers, which in the majority of cases exceeds the number of 2 needed for the Fisher’s test), while for the statistical assessments we combine various numbers to have 2 comparison groups.  We included a bottom line with exact value of P for comparisons of each table into Tables 2 to 9 (revised manuscript).

Reviewer 2 Report

Introduction. Please revise the direct citation in the paper, no page number.  There is important describe the context of the study. However, more evidence/studies about the health information (literacy) or quality of life, for example. Will be relevant in the background.

More information about the problem are need.

Methods.  In the data collection how were anonymity and confidentiality ensured? “Every participant received a printed card copy of the questionnaire at the ICDC and filled it 121 in by him or herself or a care-giver over the period of 2 weeks of April (10-25) 2020”.

In the survey there are question that need a pre-test with patients because there is some information, such ass, “Have you ever heard about Cochrane and the work of this Organisation?” , that is not easy know among the patients.

Statistically information’s details are need, for example, what descriptive analysis was performed. What program was used? The level of significance or confidence interval (CI).

The authors reported the 2 caregivers. The answers of this caregivers were about then or the patient?

Results.  How the authors operationalized this variable “Unhealthy life-style (including smoking, alcohol consumption etc)”. In the survey the participants have several options an cloud chose more than one. The same doubt about the variable “chronic diseases”.

Please clarify this paragraph “When comparing the answer “daily use” with all other answers combined together among people aged under 60 and aged 60+, it was found that more of the older participants took medicines on daily basis, 10 vs 6, compared to 29 vs 88 among participants aged under 60 (P ~ 0,003-0,006, 171 Fisher’s exact test).”

The authors included too many tables, please reduce the number and cluster the variable according the section of survey. In table provide the p-values.

The p-values provide are not exact? Why use (p≈0,002-0,003); Provide the value of the test.

Discussion. It is appropriate for the relevance of the discussion to the results.  However, the convergence or divergences with other studies is need. Many researches were done in the Implementation of EB and health literacy, that could be add to discussion section. The authors may wish to expand the discussion to include identifying similarities and differences with previous researches.

Organization and style of presentation: OK, but the results have many tables and no information about the p-values.

Reference are relevant and a significant number published in the last 5 years. However, the authors could provide the title in English.

Good luck and thank you very much for giving me a chance to review.

Author Response

Cover / rebuttal letter

Dear editors and reviewers,

With this letter we thank you for the detailed peer-review process and helpful suggestions for improvement of our manuscript “Smart and Age-Friendly Cities in Russia: A pilot survey on attitudes, perceptions, quality of life and health information needs” by Liliya Eugenevna Ziganshina, Ekaterina V. Yudina, Liliya I. Talipova, Guzel N. Sharafutdina, and Rustem N. Khairullin.

We are uploading the revised version to the Editorial manager.

Our answers to reviewers’ comments and suggestions:

Reviewer 2

Introduction. Please revise the direct citation in the paper, no page number.  There is important describe the context of the study. However, more evidence/studies about the health information (literacy) or quality of life, for example. Will be relevant in the background.

More information about the problem are need. 

Authors’ answer:

Thank you very much for this comment. We included one more paragraph in the Introduction section to make our stated aims better substantiated:

“Direct involvement of consumers in health care with shared decision making has evolved as a new approach or concept in health care over the last decades [17, 18, 19, 10]. The language to define the new concepts and approaches in research and practice evolved and required special attention to terminology which has not yet been fully unified [20]. Implementation of shared decision making should rely on knowledge and skills in evidence-based medicine not only of health professionals but also of patients or consumers for quality and transparency of decision making [19]. Research in consumer training potential emerged recently. It was shown that training in EBM was feasible [17, 18], better and long-term implementation and further research are needed [17].”

Methods.  In the data collection how were anonymity and confidentiality ensured? “Every participant received a printed card copy of the questionnaire at the ICDC and filled it 121 in by him or herself or a care-giver over the period of 2 weeks of April (10-25) 2020”.

Authors’ answer:

Thank you very much for this comment. We modified the description of how participants were recruited (lines 113-122), included description of anonymity / confidentiality and the referral to the Ethics approval, ICDC (institution), the Informed Consent Form and the measurements to read (lines 126 to 145 of the revised manuscript):

“The original plan was to focus on ageing population only, however, at the time of the pilot, which coincided with the time of expansion of COVID-19 pandemic in Kazan, due to the scarcity of in-patients we decided not to use any specific cut-off age value to include participants into our pilot. All patients, caregivers and staff members of the ICDC, who agreed to participate in the pilot at that point of time, were included. This approach allowed us to see whether there were any differences amongst various age groups in knowledge needs, perceptions, attitudes, quality of life, contentment with current position and challenges people face in maintaining and improving their health with ageing. Every participant gave his/her informed consent to participate in the pilot and to anonymized processing of personal demographic information (for the English translation of the Informed Consent form please see Appendix S2 and the Ethics approval section) before taking part in the survey and received a printed hard copy of the questionnaire at the ICDC and filled it in by him or herself or a care-giver over the period of 2 weeks of April (10-25) 2020. A total of 134 people took part in the study.

The survey results were combined in Excel spreadsheet by calculating absolute and relative numerical measurements for which we used numbers of participants of various categories (age, gender, health profession, etc) as the measurements. The results of open-ended questions were used in descriptive way. Various subgroups of participants were compared with each other to explore potential associations of studied parameters (answers to survey questions) with age, gender, representing health profession or not and other participant characteristics (but not a participant status with the ICDC (employee / patient / caregiver) using Fisher’s exact test.”

Here is the text of the Ethics approval subsection (lines 559 to 561 of the revised manuscript):

“Ethics approval: the study obtained ethics approval from the Ethics Committee of the ICDC, numbered 259 dated 30th of March 2020. All the participants gave their informed consent before taking part in the survey. The English translation of the Informed Consent form is in Appendix S2.”

In the survey there are question that need a pre-test with patients because there is some information, such ass, “Have you ever heard about Cochrane and the work of this Organisation?” , that is not easy know among the patients.

Authors’ answer:

Thank you very much for this comment. Indeed, understanding the background level of awareness / knowledge prior to our Cochrane intervention was the main objective of our pilot study (lines 108 to 111 of the revised manuscript):

“The aim of the study was to explore the health information needs of ageing population of Kazan and the challenges people face in improving their health and longevity based on their attitudes, perceptions and assessment of their quality of life in Kazan to ultimately raise evidence-based awareness about ageing.”

Statistically information’s details are need, for example, what descriptive analysis was performed. What program was used? The level of significance or confidence interval (CI).

Authors’ answer:

Thank you very much for this comment. We reworded the description of data procession to accommodate for requested details to read (there was no need in calculating any confidence intervals in view of specifics of the use the Fisher’s exact test) (lines 139 to 145 of the original manuscript): 

“The survey results were combined in Excel spreadsheet by calculating absolute and relative numerical measurements for which we used numbers of participants of various categories (age, gender, health profession, etc) as the measurements. The results of open-ended questions were used in descriptive way. Various subgroups of participants were compared with each other to explore potential associations of studied parameters (answers to survey questions) with age, gender, representing health profession or not and other participant characteristics (but not a participant status with the ICDC (employee / patient / caregiver) using Fisher’s exact test.”

The authors reported the 2 caregivers. The answers of this caregivers were about then or the patient?

Authors’ answer:

Thank you very much for this comment. Yes, the answers of the caregivers were about themselves. Since we did not compare staff with patients and with caregivers, we did not exclude caregivers from the data analysis. We presented these data (the number of caregivers) to describe participants in general, as a characteristic of the participants in relation to the ICDC (staff, patients, caregivers). But since some of the staff were of non-medical profession, while some of the patients were of medical profession, we decided that it would be important to compare the answers of people representing health/medical profession versus the answers of people not belonging to health/medical profession, and not depending on the status at the ICDC (employee / patient / caregiver). In other words we did not single out those two caregivers in our analyses.  We reworded the description of data procession to make this crystal clear (lines 139 to 145 of the original manuscript):

“The survey results were combined in Excel spreadsheet by calculating absolute and relative numerical measurements for which we used numbers of participants of various categories (age, gender, health profession, etc) as the measurements. The results of open-ended questions were used in descriptive way. Various subgroups of participants were compared with each other to explore potential associations of studied parameters (answers to survey questions) with age, gender, representing health profession or not and other participant characteristics (but not a participant status with the ICDC (employee / patient / caregiver) using Fisher’s exact test.”

Results.  How the authors operationalized this variable “Unhealthy life-style (including smoking, alcohol consumption etc)”. In the survey the participants have several options an cloud chose more than one. The same doubt about the variable “chronic diseases”.

Authors’ answer:

Thank you very much for this comment. We agree that the variable “unhealthy lifestyle” or “chronic diseases” are rather broad. However, we think that operationalising by Yes/No (having any features of unhealthy lifestyle or not; and having a chronic disease or chronic diseases or not) allowed for accurate participant categorisation, particularly given the small number of participants in our pilot study. 

Please clarify this paragraph “When comparing the answer “daily use” with all other answers combined together among people aged under 60 and aged 60+, it was found that more of the older participants took medicines on daily basis, 10 vs 6, compared to 29 vs 88 among participants aged under 60 (P ~ 0,003-0,006, 171 Fisher’s exact test).”

Authors’ answer:

Thank you very much for this comment. We agree that the provided original description required extra effort for reading through the numbers. We reworded the sentence accordingly to read (lines 217 to 220 of the revised manuscript):

“When comparing the answer “daily use” with all other answers combined together among people aged under 60 and aged 60+, it was found that more of the older participants took medicines on daily basis: 10 out of 16 compared to 29 out of 117 of people under 60 (P =0,006, Fisher’s exact test).”

The authors included too many tables, please reduce the number and cluster the variable according the section of survey. In table provide the p-values.

Authors’ answer:

Thank you very much for this comment. Indeed, we included 10 tables with analyses of survey results. However, all the tables with analyses of survey results follow the logic of the narration, are placed consistently directly below the text referring to the tables, which makes the reading straightforward and very clear. We believe that combining the tables together will lose this clarity.   

The p-values provide are not exact? Why use (p≈0,002-0,003); Provide the value of the test.

Authors’ answer:

Thank you very much for this comment. We included a bottom line with exact value of P for comparisons of each table into Tables 2 to 9 (revised manuscript) and for Tables 10 and 11 provided exact P values in the text.

Discussion. It is appropriate for the relevance of the discussion to the results.  However, the convergence or divergences with other studies is need. Many researches were done in the Implementation of EB and health literacy, that could be add to discussion section. The authors may wish to expand the discussion to include identifying similarities and differences with previous researches.

Authors’ answer:

Thank you very much for this comment. We added one more paragraph in the Introduction section to make our stated aims better substantiated. We think that at this point, when we have completed only the initial (background) survey of our planned project of Cochrane Evidence School, it would be prematurely to discuss our pilot findings in the context of other studies which looked at the impact of training. We focused the discussion on the topic of the Special issue – age-friendly cities.    

Organization and style of presentation: OK, but the results have many tables and no information about the p-values.

Authors’ answer:

Thank you very much for this comment. Indeed, we included 10 tables with analyses of survey results. However, all the tables with analyses of survey results follow the logic of the narration, are placed consistently directly below the text referring to the tables, which makes the reading straightforward and very clear. We believe that combining the table together will lose this clarity.   

Reference are relevant and a significant number published in the last 5 years. However, the authors could provide the title in English.

Authors’ answer:

Thank you very much for this comment. We translated all Russian references into English and marked them (in Russ.)

Good luck and thank you very much for giving me a chance to review.

Round 2

Reviewer 1 Report

My comments were addressed, but the merit of publishing these data and results are still questionable. It is not persuasive to publish a self-reported survey with 134 participants of mixed backgrounds and a wide age range without any novel variables measured.

Author Response

Cover / rebuttal letter (2)

Dear editors and reviewers,

With this letter we thank you for the detailed peer-review process and helpful suggestions for improvement of our manuscript “Smart and Age-Friendly Cities in Russia: A pilot survey on attitudes, perceptions, quality of life and health information needs” by Liliya Eugenevna Ziganshina, Ekaterina V. Yudina, Liliya I. Talipova, Guzel N. Sharafutdina, and Rustem N. Khairullin.

We are uploading the revised (2) version to the Editorial manager.

Our answers to reviewers’ comments and suggestions:

Reviewer 1 (2)

My comments were addressed, but the merit of publishing these data and results are still questionable. It is not persuasive to publish a self-reported survey with 134 participants of mixed backgrounds and a wide age range without any novel variables measured.

Authors’ answer:

Thank you very much for appreciating our work on the comments of the first round of revision, and for this comment.

In line with the original advice of the other reviewer (positive) that the conclusions might be improved we introduced a few additions to the Conclusions text to explicitly match the objectives of our pilot (exploratory) study to read:

“5. Conclusions

This pilot study provides data on health information needs of ageing and middle-aged population of Kazan by documenting that less than half of the survey participants were aware of evidence-based medicine and only quarter of them knew about Cochrane and Cochrane work. EBM-awareness was higher within the healthcare profession than others and among men than women with a weak tendency to be higher in participants under 60 years of age than in people of 60+ years old. Cochrane awareness was also higher within the healthcare profession. The need for EBM and Cochrane knowledge was also higher within the health profession, without age or gender differences, or between participants with satisfactory and unsatisfactory quality of life.

Most of the survey participants were generally positive in assessing their quality of life, despite the fact that many have long-term health conditions and unhealthy lifestyles. A few reported improvements of quality of life with age. 

Older participants were less positive about their quality of life. More participants aged 60+ took medicines on a daily basis than people under 60 years of age.

Ageing was the challenge in improving health and longevity with more participants aged 60+ rating their quality of life as unsatisfactory than participants under 60 years old, and encountering more age-related problems. Only ten percent of participants were aware of ageism without age or gender differences.

Low awareness of EBM, Cochrane, ageing problems and ageism together with the scarcity of independent accessible quality information about effectiveness and safety of therapeutic and diagnostic interventions, non-drug treatment and prevention measures argue for the urgent need for raising awareness of ageing problems, ageism and evidence-based approach to tackling the problems of ageing population of Kazan, which strives to be an age-friendly city.

Cochrane Evidence Schools designed not only for patients and consumers, but also for health care providers might become a beneficial instrument both for raising awareness and advancing knowledge and attitudes of the citizens of Kazan as a pilot city with potential of scaling up.

The hope is that this pilot study would contribute to Kazan becoming not only a fully age-friendly city, but moving towards age-inclusiveness.”

We disagree that the merit of publishing is questionable, because we did measure novel variables, in fact all measured variables are absolutely new for the Russian health and societal landscape.

We did not pretend that we were doing epidemiological study, requiring large numbers of units of observation. The number of participants allowed us to detect a few pilot associations which we plan to explore further when progressing with the project and scaling up our studies. 

Reviewer 2 Report

Dears Authors

Thanks for your revisions I have no further comments

Author Response

Cover / rebuttal letter (2)

Dear editors and reviewers,

With this letter we thank you for the detailed peer-review process and helpful suggestions for improvement of our manuscript “Smart and Age-Friendly Cities in Russia: A pilot survey on attitudes, perceptions, quality of life and health information needs” by Liliya Eugenevna Ziganshina, Ekaterina V. Yudina, Liliya I. Talipova, Guzel N. Sharafutdina, and Rustem N. Khairullin.

We are uploading the revised (2) version to the Editorial manager.

Our answers to reviewers’ comments and suggestions:

Reviewer 2 (2)

Dears Authors

Thanks for your revisions I have no further comments

Authors’ answer:

Dear Reviewer,

Thank you very much for all your comments and suggestions, which contributed a lot to the substantial improvement of our manuscript.

In line with your advice that the conclusions might be improved we introduced a few additions to the Conclusions text to explicitly match the objectives of our pilot (exploratory) study to read:

“5. Conclusions

This pilot study provides data on health information needs of ageing and middle-aged population of Kazan by documenting that less than half of the survey participants were aware of evidence-based medicine and only quarter of them knew about Cochrane and Cochrane work. EBM-awareness was higher within the healthcare profession than others and among men than women with a weak tendency to be higher in participants under 60 years of age than in people of 60+ years old. Cochrane awareness was also higher within the healthcare profession. The need for EBM and Cochrane knowledge was also higher within the health profession, without age or gender differences, or between participants with satisfactory and unsatisfactory quality of life.

Most of the survey participants were generally positive in assessing their quality of life, despite the fact that many have long-term health conditions and unhealthy lifestyles. A few reported improvements of quality of life with age. 

Older participants were less positive about their quality of life. More participants aged 60+ took medicines on a daily basis than people under 60 years of age.

Ageing was the challenge in improving health and longevity with more participants aged 60+ rating their quality of life as unsatisfactory than participants under 60 years old, and encountering more age-related problems. Only ten percent of participants were aware of ageism without age or gender differences.

Low awareness of EBM, Cochrane, ageing problems and ageism together with the scarcity of independent accessible quality information about effectiveness and safety of therapeutic and diagnostic interventions, non-drug treatment and prevention measures argue for the urgent need for raising awareness of ageing problems, ageism and evidence-based approach to tackling the problems of ageing population of Kazan, which strives to be an age-friendly city.

Cochrane Evidence Schools designed not only for patients and consumers, but also for health care providers might become a beneficial instrument both for raising awareness and advancing knowledge and attitudes of the citizens of Kazan as a pilot city with potential of scaling up.

The hope is that this pilot study would contribute to Kazan becoming not only a fully age-friendly city, but moving towards age-inclusiveness.”

Thank you very much indeed for your positive and very helpful review.
